# Aurintricarboxylic acid is a canonical disruptor of the TAZ-TEAD transcriptional complex

**Kepeng Che[1], Ajaybabu V. Pobbati[1], Caleb N. Seavey[1,2], Yuriy Fedorov[3], Anton A. Komar[4,5], Ashley Burtscher[1], Shuang Ma[1], Brian P. Rubin[1,6]***

**1** Department of Cancer Biology, Lerner Research Institute, Cleveland Clinic, Cleveland, Ohio, United States of America, **2** Department of General Surgery, Digestive Disease and Surgery Institute, Cleveland Clinic, Cleveland, Ohio, United States of America, **3** Small Molecule Drug Development Core, Case Western Reserve University School of Medicine, Cleveland, Ohio, United States of America, **4** Department of Biological, Geological and Environmental Sciences, Center for Gene Regulation in Health and Disease, Cleveland State University, Cleveland, Ohio, United States of America, **5** Genomic Medicine Institute, Lerner Research Institute, Cleveland Clinic, Cleveland, Ohio, United States of America, **6** Robert J. Tomsich Pathology and Laboratory Medicine Institute, Cleveland Clinic, Cleveland, Ohio, United States of America

* rubinb2@ccf.org

**Data Availability Statement:** Raw RNA-Seq data is deposited in GEO database at accession number GSE198108. All other relevant data are within the manuscript and supporting information files.

## Abstract

Disrupting the formation of the oncogenic YAP/TAZ-TEAD transcriptional complex holds substantial therapeutic potential. However, the three protein interaction interfaces of this complex cannot be easily disrupted using small molecules. Here, we report that the pharmacologically active small molecule aurintricarboxylic acid (ATA) acts as a disruptor of the TAZ-TEAD complex. ATA was identified in a high-throughput screen using a TAZ-TEAD AlphaLISA assay that was tailored to identify disruptors of this transcriptional complex. We further used fluorescence polarization assays both to confirm disruption of the TAZ-TEAD complex and to demonstrate that ATA binds to interface 3. We have previously shown that cell-based models that express the oncogenic TAZ-CAMTA1 (TC) fusion protein display enhanced TEAD transcriptional activity because TC functions as an activated form of TAZ. Utilizing cell-based studies and our TC model system, we performed TC/TEAD reporter, RNA-Seq, and qPCR assays and found that ATA inhibits TC/TEAD transcriptional activity. Further, disruption of TC/TEAD and TAZ/TEAD interaction by ATA abrogated anchorage-independent growth, the phenotype most closely linked to dysregulated TAZ/TEAD activity. Therefore, this study demonstrates that ATA is a novel small molecule that has the ability to disrupt the undruggable TAZ-TEAD interface.

## Introduction

The transcriptional coactivators, YAP (Yes-associated protein) and TAZ (transcriptional coactivator with PDZ-binding motif) play key roles in cancer initiation, progression, and drug resistance [1–3]. The transcriptional activities of YAP and TAZ are physiologically kept under the tight control by the Hippo signaling pathway. In cancers, the Hippo signaling components are either inactivated by loss-of-function mutations or epigenetically silenced, leading to

**Funding:** This research is supported by generous funding from the EHE Foundation, The EHE Rare Cancer Charity (UK), The EHE Rare Cancer Foundation Australia and the Margie and Robert E. Petersen Foundation awarded to B.P.R. These are Foundation grants and they do not carry a grant number. Screening campaign was supported by the Small Molecule Drug Development shared resource of the Case Comprehensive Cancer Center (P30CA043703). C.N.S. is supported in part by the Crile Research Fellowship. The funders had no role in study design, data collection and analysis, decision to publish, or preparation of the manuscript.

**Competing interests:** The authors have declared that no competing interests exist.

enhanced YAP/TAZ activities [4]. Alternatively, cancer cells have also been shown to overwhelm Hippo signaling through YAP and TAZ copy number increases, gain-of-function mutations, or through the activity of YAP/TAZ fusion genes, all of which enhance YAP/TAZ transcriptional activity. Additionally, tissue architectural changes and cues from a stiff extracellular matrix present a hospitable tumor microenvironment that also allows for enhanced YAP/TAZ activities [5].

Regardless of these activating mechanisms, YAP and TAZ need to pair with transcription factors to access DNA [6, 7]. TEADs (TEA/ATTS domain), a family of four highly conserved transcription factors, have emerged as the major player that orchestrates several oncogenic transcriptional programs through interacting with YAP and TAZ. TEADs interact with YAP/TAZ through a C-terminal YAP/TAZ-binding domain that adopts a β-sandwich fold [8–11]. Genetic and cell-based studies have shown that disruption of a functional YAP/TAZ-TEAD interaction greatly reduces oncogenicity. For instance, a transgenic mouse model that expresses a dominant negative TEAD, TEAD without its DNA-binding domain, potently inhibits liver tumorigenesis [12]. A splicing switch that activates a similar dominant negative TEAD isoform also acts as an inhibitor of tumorigenesis [13]. Further, YAP is no longer oncogenic if it carries a point mutation that compromises its ability to interact with TEADs [14]. Therefore, disruption of the YAP/TAZ interaction with TEAD is being actively investigated as a strategy for cancer therapy [15].

Unlike YAP/TAZ, TEADs are targetable as they have pockets that accommodate small molecule and peptide-based targeting strategies [16, 17]. TEADs have three distinct pockets that can be leveraged to disrupt the YAP/TAZ-TEAD interaction–a pocket at the center of the YAP/TAZ-binding domain (central pocket) and two surface pockets where YAP/TAZ interact (interfaces 2 and 3) [16]. The central pocket is more druggable, so identifying small molecule inhibitors that act at the central pocket is much easier than identifying canonical disruptors that act at the YAP/TAZ-TEAD interaction interfaces on the surface [18]. However, central pocket cysteines are acylated with palmitate or myristate, and these covalent ligands block small molecule access [19–21]. On the other hand, small molecules occupying the surface pockets have the potential to disrupt the YAP/TAZ-TEAD interaction [15, 16]. Because the surface pockets are shallow, identifying a small molecule that is potent enough to disrupt this protein-protein interaction remains a challenge.

Designing novel and highly active YAP/TAZ-TEAD disruptors is an emerging field, several chemotypes must be identified and evaluated, so a clinically-relevant drug candidate can be developed. To this end, we have designed a TAZ-TEAD AlphaLISA assay, a biochemical assay that monitors the formation of the TAZ-TEAD complex. We screened compound libraries totaling 56,115 compounds and identified aurintricarboxylic acid (ATA) as a small molecule drug that binds to the TEAD surface and disrupts the formation of the TAZ-TEAD complex. Our specific clinical focus is on epithelioid hemangioendothelioma (EHE), a rare cancer that is addicted to the TAZ-CAMTA1 (TC) fusion protein that functions as an activated form of TAZ [22, 23]. Importantly, EHE provides a model for dysregulation of YAP/TAZ activity in cancer, and conclusions generated from investigations on EHE are applicable to other cancers in which YAP/TAZ is dysregulated. We have previously demonstrated that TC-mediated cell transformation is TEAD-dependent, as the TEAD binding motif is maintained in the fusion protein, and abrogation of the TC/TEAD interaction completely inhibits cell transformation [23]. Here, we show that ATA can abolish TC/TEAD-mediated transcription and TC/TEAD-dependent transformation in our cell-based models. These results demonstrate that ATA operates as a TC/TEAD disruptor in both biochemical and cell-based models.

## Materials and methods

### Reagents

Reagents purchased were ATA (Sigma-Aldrich, catalog # A1895), anti-His (Abcam, catalog # Ab18184), anti-FLAG (Sigma-Aldrich, catalog # F1804), Protein Thermal Shift dye (Applied Biosystems, catalog # 4461141), Ni-NTA agarose beads (Qiagen, catalog # 30210), strep-tactin superflow resin (Qiagen, catalog # 30004), verteporfin (Sigma-Aldrich, catalog # SML0534), and Peptide 17 (Selleckchem, catalog # S8164).

### Protein expression and purification

The cDNAs of human TEAD4 (1–434) (UniProt—Q15561) and TAZ (UniProt—Q9GZV5) were codon-optimized according to a proprietary algorithm developed by DAPCEL, Inc [24], synthesized, and inserted into the pET-3a vector between restriction sites NdeI and BamHI (GenScript). The final construct contained a N-terminal strep-TEV-FLAG tag for TEAD4, a 6x His tag at the N-terminus and a strep tag at the C-terminus of TAZ (GenScript).

For protein expression, the strep-TEV-FLAG-TEAD4 plasmid and His-TAZ-strep plasmid were expressed in *Escherichia coli* strain BL21(DE3)pLysS competent cells (Invitrogen). The strep-tagged TEAD4 was batch purified using strep-tactin superflow resin. TEV protease was added to remove the strep tag. The His-TAZ-strep protein was first passed through Ni-NTA beads, and then was further purified using strep-tactin superflow resin. Protein purity was analyzed by SDS-PAGE and western blots (S1A Fig). The YAP/TAZ-binding domain of human TEAD4 (217–434) was purified using immobilized metal affinity chromatography followed by size exclusion chromatography.

### TAZ-TEAD AlphaLISA assay development and high-throughput screening

Full-length TAZ and TEAD proteins were used in this assay. The codon-optimized cDNAs yielded sufficient protein amounts to carry out the high-throughput screen (S1A Fig).

Purified full-length His-TAZ and FLAG-TEAD4 proteins were mixed with nickel chelate AlphaLISA Acceptor Beads and anti-FLAG Alpha Donor Beads (PerkinElmer) in alpha assay buffer (10mM Tris pH 8.0, 150mM NaCl, 0.1% BSA, 0.01% Tween 20, 10 mM DTT). To determine the optimum protein concentration that produced the highest total signal intensity, His-TAZ was used at a concentration range of 0 nM to 20 nM and FLAG-TEAD4 at 0.1 nM to 1000 nM. The maximal signal occurred at 10 nM TAZ and 15 nM TEAD4 (S1B Fig). Then, compounds from the library plates were added at a final concentration of 10 μM. The mixtures were incubated in the dark for 2h at room temperature and emission at 615 nm was measured using the alpha-compatible EnSpire (PerkinElmer) and Synergy Neo2 (BioTek) multimode plate readers.

In the high-throughput screen (HTS), a library of 53,000 compounds with drug-like physicochemical properties (DIVERset-CL, ChemBridge) was used. Stock solutions were prepared in dimethyl sulfoxide (DMSO) at 10 mM. All DIVERset-CL collection compounds were screened at 10 μM. Additionally, we used two bioactive libraries with a total of 3115 small molecules compiled from the Sigma LOPAC1280 (1280 molecules) library and the Selleck Bioactives L1700 library (1835 molecules). For both bioactive libraries, stock solutions were prepared in DMSO at 3 mM. All bioactives collections compounds were screened at 3 μM. HTS is performed in a singlicate.

Selected hits were re-tested with a counterscreen. The peptide used for the counterscreen was 24 amino acids long and contained a 6X His tag at the N-terminus and a FLAG-tag at the

C-terminus with a linker region separating the tags (Genscript). This peptide brings the donor and acceptor beads close enough to facilitate singlet oxygen transfer.

## ThermoFluor assay

To assess changes in protein stability, a dye-based thermal melting assay (ThermoFluor) was used. Purified protein (TAZ or TEAD4), ATA, and Protein Thermal Shift dye (ThermoFisher) were mixed in 96-well plates. The final concentration of the protein was 5 μM. The melting temperature was measured using LightCycler 480 (Roche). Each experiment was repeated at least twice.

## Surface plasmon resonance

The YAP/TAZ-binding domain of human TEAD4 (217–434) was deacylated [25] and resuspended in 10 mM acetate buffer pH 5.0. TEAD was immobilized on a CM5 sensor chip (Cytiva, product, BR100530) using the amine coupling kit (Cytiva, product, BR10050) as per manufacturer's guidelines. Various concentrations of ATA were dissolved in buffer containing 20 mM Tris pH 8.0, 150 mM NaCl, 1.5% DMSO and passed over the reference cell and the cell containing immobilized TEAD, and the resonance responses were recorded.

## Fluorescence polarization assays

Human YAP (61–100), human TAZ (23–57) and mouse Vgll1 (26–51) peptide probes were used; all the probes had carboxyfluorescein labels. Human TEAD4 (217–434) was used in the assay at a concentration of 350 nM and the peptide probe concentration was 25 nM. Disruptors were titrated at concentrations ranging from 0–150 μM to the probe-TEAD complex, and the mixtures were incubated at 25˚C for 20 min. Then, polarization measurements were taken using a Victor3 (PerkinElmer) plate reader.

## Cellular assays

NIH3T3 and HEK293 cells were obtained from the American Type Tissue Collection (ATCC, Manassas, VA) and cultured in DMEM containing 10% fetal bovine serum and penicillin/streptomycin. A TEAD dual luciferase reporter assay was performed as previously described [23]. Cells were treated with vehicle control or ATA (10 μM) for 48 h before measuring the luciferase activity or processing the samples for RNA extraction. Each experiment was repeated at least 2–3 times.

## Bulk RNA-Seq and qPCR experiments

NIH3T3 cells were used. The extracted RNA (RNeasy Mini, Qiagen) from treated cells was either used for qPCR measurements or sent for bulk RNA-Seq. Sequencing runs were performed on an Illumina HiSeq 4000 sequencer (Genewiz). Subsequent bioinformatics analysis was performed in the Galaxy Project platform [26]. Paired end reads were aligned to the murine genome (GRCm38.p6 mm10) with HISAT2 [27]. Raw count files were generated with featureCounts using a corresponding GTF reference [28]. Differential gene expression analysis was performed with DESeq2 [29]. Transcripts were considered to be differentially expressed if the false discovery rate $< 0.05$ and $Log_2FC > 2$. When performing differential gene expression for GSEA analysis, the Limma package was used ($<5$ CPM filtering) with t-scores used for rank statistics [30]. GSEA plots and statistics were generated with FGSEA with custom gene sets. Raw sequencing reads and processed files have been deposited in the GEO repository

(data will be uploaded upon manuscript acceptance). The Universal Probe Library probes (Roche) and the associated primer sequences for the qPCR targets are listed below:

*Ccn1/Ctgf* F: TGACCTGGAGGAAAACATTAAGA, *Ccn1/Ctgf* R: AGCCCTGTATGTCTTCAC ACTG,

*Ccn1/Ctgf* UPL Probe: 71;

*Ccn2/Cyr61* F: GGATCTGTGAAGTGCGTCCT, *Ccn2/ Cyr61* R: CTGCATTTCTTGCCCTTT TT,

*Ccn2/Cyr61* UPL Probe: 66;

*Ankrd1* F: GCTGGAGC  CCAGATTGAA, *Ankrd1* R: CTCCACGACATGCCCAGT,

*Ankrd1* UPL Probe: 76;

*Amotl2* F: TGACTGTACCTAAGCCGAACC; *Amotl2* R: GCACACACCTGCCTAGACAAT,

*Amotl2* UPL Probe: 40;

*Gapdh* F: CCCACTTGAAGGGTGGAG, *Gapdh* R: TGGTTCACACCCATCACAAA,

*Gapdh* UPL Probe: 29.

## Soft agar colony growth assay

Soft agar colony growth assay was performed as previously described [23]. The colonies were counted using Image-Pro Plus 7.0 software (Media Cybernetics). ATA cytotoxicity was assessed using the CyQUANT LDH Cytotoxicity Assay Kit (ThermoFisher) and was performed as per the manufacturer's protocol.

# Results

## TAZ-TEAD AlphaLISA assay and high-throughput screening

We designed the TAZ-TEAD AlphaLISA assay to identify small molecules that possess the ability to disrupt the formation of the TAZ-TEAD complex (Fig 1A). It is also suitable for screening compounds in a high-throughput format. His-TAZ interaction with FLAG-TEAD4 was optimized such that donor and acceptor beads were close enough to enable the singlet oxygen generated from donor beads to react with the acceptor AlphaLISA beads causing a chemiluminescent emission at 615 nm (Fig 1A). This optimized AlphaLISA assay had a signal-to-background ratio range of 35 to 40. Small molecule disruptors that abrogate the TAZ-TEAD interaction should cause a reduction in the AlphaLISA signal.

Utilizing our optimized assay, we screened small molecule libraries, totaling 56,115 compounds. The overall Z' for the assay is 0.7, suggesting that the assay is of good quality. We identified 287 hits that reduced by 70% the AlphaLISA signal increase caused by TAZ-TEAD interaction (S1C Fig). We subsequently performed a counterscreen to identify non-specific assay inhibitors. The counterscreen showed that the majority of hits were assay-interfering compounds (Fig 1B). No interference in the counterscreen was observed in 5 hits, indicating that these hits specifically reduced the TAZ-TEAD AlphaLISA signal. These 5 hits also showed a dose-dependent reduction in the TAZ-TEAD AlphaLISA assay signal, but not in the counterscreen. The structure of one of the hits, aurintricarboxylic acid (ATA), and its dose-response curves are shown in Fig 1C and 1D, respectively. Next, we used a dye-based thermal shift (ThermoFluor) assay that measures the melting temperature ($T_m$) of the protein to determine whether the shortlisted hits bind to TAZ or TEAD. Only ATA increased the $T_m$, and stabilized TEAD more than TAZ in the ThermoFluor assay, suggesting that it binds to TEAD, as opposed to TAZ (Fig 1E and 1F). We focused on ATA because the other 4 hits displayed no shift in $T_m$.

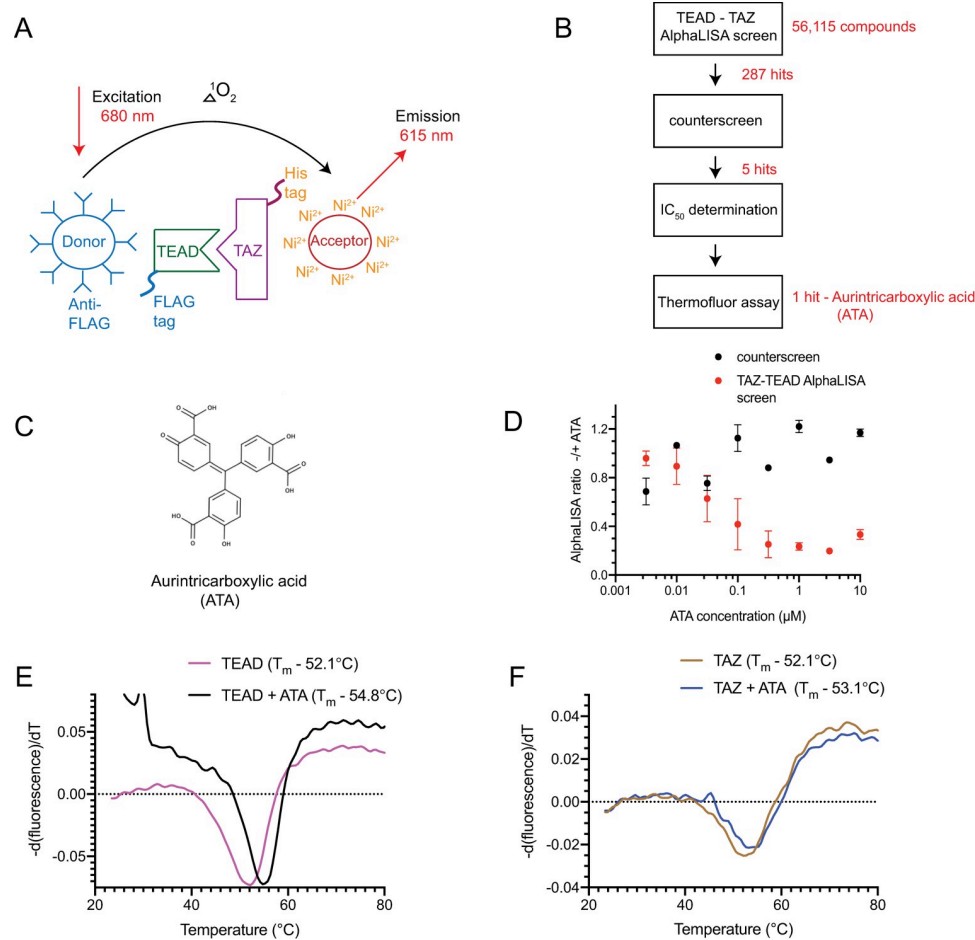

**Fig 1. TAZ-TEAD AlphaLISA assay. A,** A schematic showing the TAZ-TEAD AlphaLISA assay design. The interaction between the tagged, full-length, TAZ and TEAD proteins brings the donor and acceptor beads close enough to allow singlet oxygen transfer, which generates an emission signal at 615 nm. **B,** Flowchart of the screening strategy, 56,115 compounds were screened using the TAZ-TEAD AlphaLISA screen. Filtering the hits via a counterscreen followed by $IC_{50}$ determination and assessment using a ThermoFluor assay identified aurintricarboxylic acid (ATA) as a disruptor of the TAZ-TEAD complex. **C,** The molecular structure of ATA. **D,** Dose-response curves obtained after titration of ATA either in the TAZ-TEAD AlphaLISA assay or in the counterscreen; a FLAG-His fusion peptide was used in the counterscreen. **E and F,** ThermoFluor assays measuring the melting temperatures ($T_m$) of full-length TAZ or TEAD in the presence of ATA. Data are representative of three independent experiments performed using technical duplicates.

## Synthesis of ATA analogs

A minimum pharmacophore study was conducted to identify key functionalities of ATA that disrupt the TAZ-TEAD interaction. In total, 16 ATA analogs were tested. Compounds 1 and 2, which retained three phenyl substituents like ATA but lacked the salicylate group, were not appreciably potent (S1 Table). Therefore, the salicylate group is important for disrupting the TAZ-TEAD interaction. To further probe the requirement of the salicylate group, we assayed five analogs (compounds 3, 4, 5, 6, and 7) that had two aryl substituents. Compound 4, which featured two salicylate moieties, was the most potent, with an $IC_{50}$ of 8 μM in the TAZ-TEAD AlphaLISA assay (S1 Table). Although compound 4 displayed activity, it was not as effective as ATA, which had a third aryl substituent in addition to the two salicylate moieties. Therefore, three aryl substitutions, including two salicylate moieties may be important for effective

disruption of TAZ-TEAD interaction. Additionally, no analogs with one or zero aryl substituents displayed inhibition. Next, we evaluated five derivatives of compound 4 by introducing mono-, di- or gem-dimethyl groups at the methylene linker, or capped hydroxyls with methyl groups, and assessed their potencies (compounds 12–16, S1 Table). Compound 14 was estimated to be most potent of the 16 ATA analogs with an $IC_{50}$ of 4 μM. This compound had a mono-methyl group at the methylene linker. We also observed that capping the hydroxyls with methyl groups as in compound 16 abolished the activity. Nevertheless, all the tested analogs were substantially less potent than ATA, which had an $IC_{50}$ of 35 nM (S1D Fig). As a result, we further characterized ATA and not its analogs in subsequent studies.

## ATA binds to TEAD and disrupts the TAZ-TEAD interaction

We performed surface plasmon resonance experiments to verify whether ATA binds to TEAD. The YAP/TAZ-binding domain of TEAD4 was immobilized on the sensor chip and resonance responses were recorded after passing various concentrations of ATA through the flow cell (Fig 2A). The affinity of the interaction was measured after fitting a one-site binding model to a plot of steady-state response versus concentration data (Fig 2B). Our results show that ATA binds to TEAD in the low micromolar range (Fig 2B).

To verify whether ATA is a disruptor of TAZ-TEAD interaction using another independent method, we developed a TAZ-TEAD fluorescence polarization (FP) assay. TAZ interacts with TEAD by forming interfaces 2 and 3 (Fig 2C) and interface 3 residues are key mediators of the TAZ-TEAD interaction [11]. The degree of polarization increased as TEAD formed a complex with a labeled TAZ peptide probe. Upon ATA titration, the fluorescence polarization (mP) of the TAZ-TEAD complex decreased in a dose-dependent manner and reached a value similar to that of the free TAZ probe, indicating that ATA acts as a disruptor of the TAZ-TEAD complex (Fig 2D). We also investigated whether ATA acts as a canonical disruptor (i.e., a disruptor that binds at the interface where YAP/TAZ interacts with TEAD), or an allosteric disruptor (by binding to the central pocket). Small molecules have also been shown to occupy the central pocket and allosterically disrupt the formation of the YAP/TAZ-TEAD complex [16]. Allosteric disruptors do not function when TEAD is acylated because the covalently attached lipid blocks small molecules access to the central pocket. Therefore, we used acylated and deacylated forms of TEAD in the TAZ-TEAD FP assay, and ATA displayed similar $IC_{50}$ values in both circumstances (Fig 2D). This result suggests that ATA binds to the TEAD surface pockets either at interface 2 or interface 3 and functions as a canonical disruptor of the TAZ-TEAD interaction. The YAP-TEAD interaction was similarly investigated using a YAP-TEAD FP assay where a YAP peptide probe was used instead of a TAZ probe. The TEAD-interacting motifs of YAP and TAZ are highly similar, barring the linker region that connects the interface 2 and 3 residues. Interestingly, ATA more potently disrupts the TAZ-TEAD complex, as compared to the YAP-TEAD complex ($IC_{50}$: 72μM vs. 160μM) (Fig 2D and 2E).

We optimized another fluorescence polarization assay, a VGLL1-TEAD FP assay, to delineate whether ATA binds at interface 2 or interface 3. VGLL1 interacts with TEAD by forming interfaces 1 and 2 [31], and not interface 3 (Fig 2C). ATA was ineffective at disrupting the VGLL1-TEAD interaction whereas peptide 7 [32] a known disruptor that acts by binding at interface 2, was highly effective. Therefore, we predict that ATA binds at interface 3. In support of this prediction, we find that the cyclic YAP peptide, peptide 17 [33], a known interface 3 binding ligand, displayed a profile similar to that of ATA in the VGLL1-TEAD FP assay (Fig 2F). Using the TAZ-TEAD FP assay, we also show a comparison of the activity profile of ATA with those of verteporfin and peptide 17, which are reported to act as disruptors (S2 Fig).

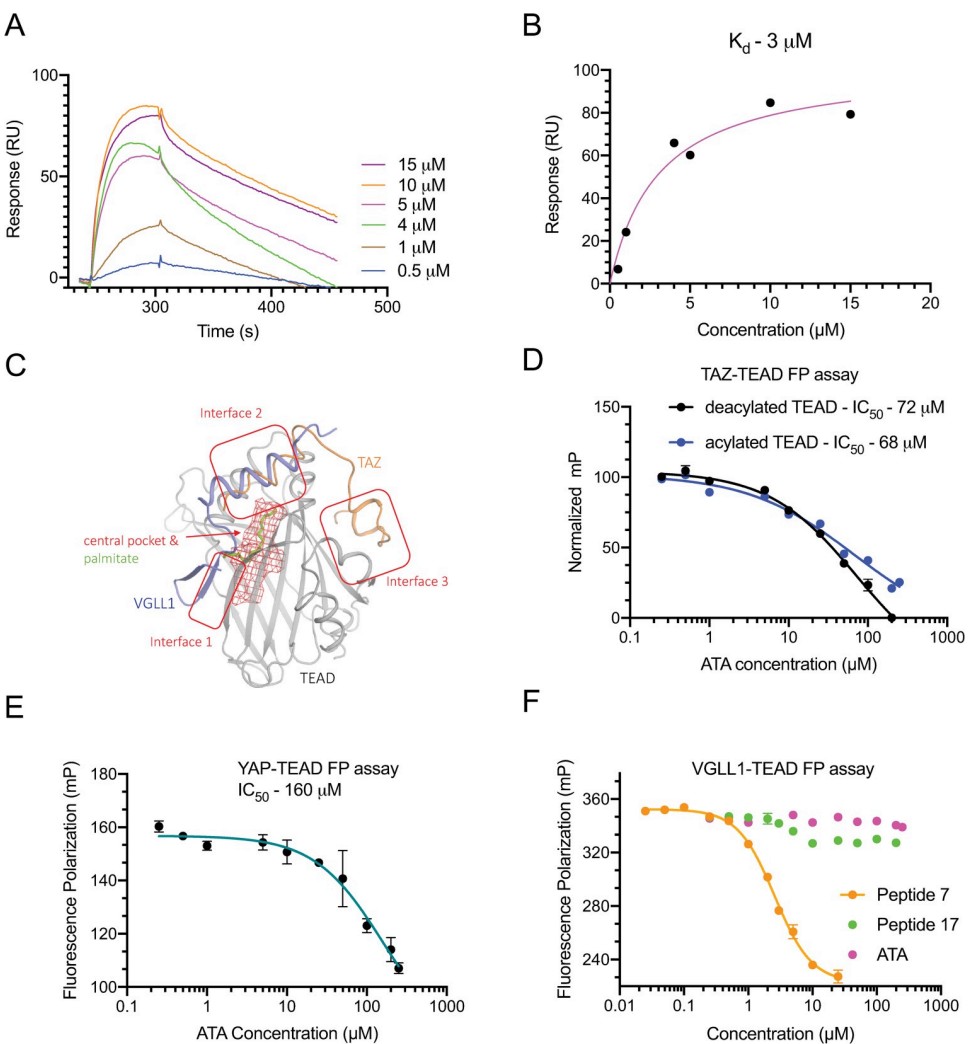

**Fig 2. ATA binds to TEAD and disrupts the TAZ-TEAD interaction. A,** Sensorgrams showing resonance responses that were recorded after passing the indicated concentrations of ATA over the YAP/TAZ-binding domain of TEAD. **B,** Affinity between TEAD and ATA was calculated by fitting a one-site binding curve to the steady-state response versus concentration. **C,** Cartoon showing the superposition of VGLL1-TEAD (PDB ID: 5Z2Q) and TAZ-TEAD structures (PDB ID: 5GN0). TAZ forms interfaces 2 and 3, whereas VGLL1 forms interfaces 1 and 2. The central pocket that houses the lipid is shown as a red mesh. **D,** Acylated and deacylated forms of TEAD were used in a TAZ-TEAD FP assay to characterize whether ATA binds to the surface or the central pocket of TEAD, the $IC_{50}$ values were obtained through a four-parameter curve fit. **E,** YAP-TEAD FP assay using a YAP peptide probe to evaluate whether ATA disrupts the formation of the YAP-TEAD complex. **F,** Vgll1-TEAD FP assay using a mouse Vgll1 peptide probe to monitor whether ATA binds at interfaces 1 and 2. Peptide 7 was used as a positive control and peptide 17 was used as a negative control. All FP experiments were repeated three times using technical duplicates with similar outcome. The data is shown as mean (n = 2) and error bars represent standard deviation of the mean.

## ATA inhibits TC/TEAD transcriptional activity

To modulate gene transcription, TEADs need to pair with transcriptional co-activators like YAP and TAZ. Therefore, we tested whether disruption of TAZ-TEAD interaction by ATA reduces TAZ/TEAD-dependent transcriptional activity. Here, we exploited the activity of the oncogenic TC fusion protein. TC is a fusion protein present in >90% of cases of a rare vascular sarcoma, epithelioid hemangioendothelioma [34, 35]. Previously, we have shown that TC acts as an activated TAZ-like protein that similarly requires the TEAD-binding motif of TAZ to

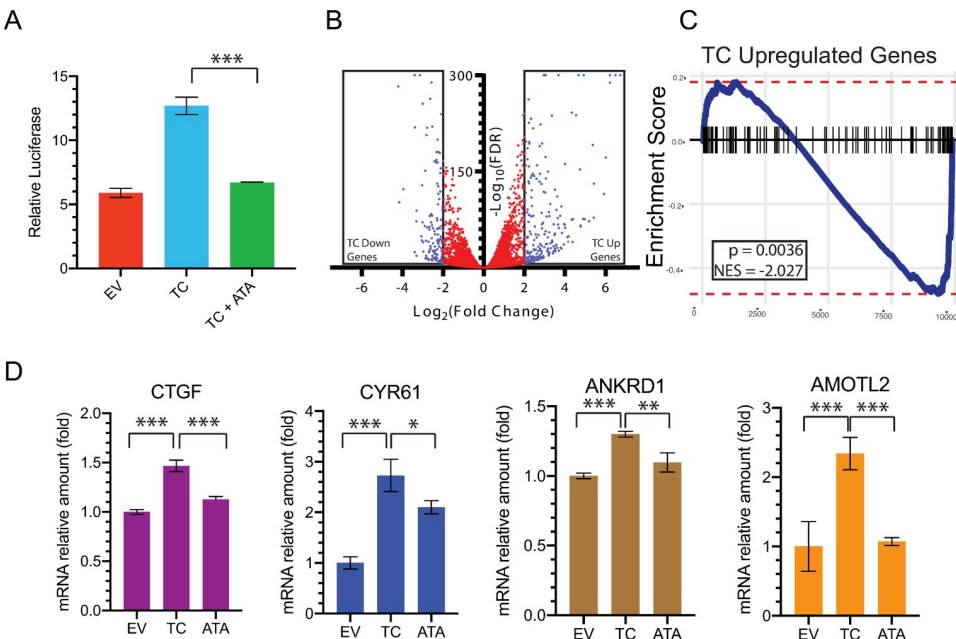

**Fig 3. ATA inhibits TC/TEAD transcriptional activity. A,** TC/TEAD reporter assay to measure TC/TEAD activity in cells transfected with empty vector (EV), cells stably expressing TC treated with vehicle control (TC), and TC-expressing cells treated with ATA (TC + ATA). The experiment was repeated three times and similar results were obtained. Data are presented as mean ± SD. **B,** Volcano plot showing the distribution of differentially expressed genes in TC-transfected compared to EV-transfected cells, blue points: FDR < 0.05 and either $Log_2FC > 2$ or $Log_2FC < -2$, red points: FDR > 0.05 or $2 < Log_2FC > -2$, FDR: False discovery rate. **C,** Gene set enrichment analysis of ATA-treated cells stably expressing TC versus TC-expressing, vehicle control treated cells utilizing the TC up Genes from Fig 3B, NES: Normalized enrichment score **D,** qPCR to probe the levels of indicated target genes in empty EV, TC and TC +ATA NIH3T3 cells. Error bars represent the standard deviation of the mean (n = 3) normalized to EV, *P* values were obtained through two-tailed *t* tests (***$P < 0.0008$, **$P < 0.008$, *$P < 0.03$).

interact with TEADs [23]. Therefore, exogenous expression of TC in NIH3T3 or HEK293 cells increases TC/TEAD-dependent transcription [23].

Using a TEAD luciferase reporter assay, we determined whether disruption of the TC/TEAD interaction in cells changed the protein complex's ability to alter gene transcription. As expected, when compared to empty vector (EV), exogenous expression of TC in HEK293 cells robustly increased luciferase reporter activity driven by TEAD DNA elements (Fig 3A). Addition of ATA significantly reduced the TC-dependent increase of the reporter activity (Fig 3A). Disruption of the TEAD interaction with TC is expected to reduce the TC/TEAD reporter activity.

We performed bulk RNA-Seq analysis in NIH3T3 cells to identify the global transcriptional alterations that are induced by TC and to probe the effect that ATA treatment has on the TC/TEAD transcriptional profile. First, differential gene expression analysis was performed between EV-transfected and TC-transfected cells to identify the TC-dependent transcriptional profile (Fig 3B). We identified 248 genes that were significantly upregulated ($Log_2FC \geq 2$, FDR $\leq$ 0.05) and 133 genes that were significantly downregulated ($Log_2FC \leq -2$, FDR $\leq$ 0.05) in TC-transfected cells, compared with EV- transfected cells (Fig 3B and S2 Table). To probe whether ATA was able to reverse the TC/TEAD transcriptional profile, we identified the differentially expressed genes in TC-expressing NIH3T3 cells after ATA treatment compared to vehicle control and performed Gene Set Enrichment Analysis (GSEA) to probe for the enrichment of TC upregulated genes (Fig 3C). ATA-treated cells displayed a negative enrichment

score for the 248 TC-upregulated genes ($P$ = 0.0036 NES = -2.027) (Fig 3C), thereby further demonstrating that ATA inhibits TC/TEAD-dependent transcription.

We further validated these results with canonical TAZ target genes (*CYR61*, *CTGF*, *AMOTL2* and *ANKRD1*) by qPCR and demonstrate a similar significant induction of these transcripts with TC and suppression of these transcripts upon ATA treatment (Fig 3D). In summary, we show that ATA can suppress the TC/TEAD-dependent gene signature in cells, thereby demonstrating that ATA-mediated disruption of the TC/TEAD interaction translates to inhibition of transcriptional activity.

## ATA specifically reduces soft agar colony growth that is dependent on TC/ TAZ-TEAD activity

We previously reported that in comparison to non-transformed NIH3T3 cells that do not grow in soft agar, expression of TC transforms these cells such that they can sustain anchorage-independent growth and form colonies in soft agar. This phenotype depends on TC/ TEAD because expression of a mutant TC that does not interact with TEAD proteins (TC [S51A]) and abrogates this interaction also abolishes colony formation [23]. Here, adding of ATA caused a dose-dependent reduction in TC-driven soft agar colony growth (Fig 4A). Likewise, ATA inhibited soft agar growth that is driven by TAZ [S89A], an activated TAZ mutant that is refractory to Hippo pathway-mediated inhibition (Fig 4B). Importantly, the observed phenotype was not due to a cytotoxic effect as ATA-treated cells did not show cytotoxicity at the concentrations used, as measured by a lactate dehydrogenase assay that measures the levels of intracellular lactate released in the media (S3 Fig). Further, as NRAS [G12V]-induced soft agar growth is unaffected by ATA treatment (Fig 4C), we infer that ATA specifically inhibits TAZ/ TEAD activity. Mutant RAS family proteins transforms cells through YAP/TAZ-independent mechanisms [36].

## Discussion

We identified ATA as a canonical protein-protein interaction disruptor of the TAZ-TEAD complex based on two independent assays, an AlphaLISA assay, and a fluorescence polarization assay. Using surface plasmon resonance and a ThermoFlour assay, we also showed that ATA directly binds to the YAP/TAZ-binding domain of TEAD. This domain has three distinct pockets: a central pocket, and two surface pockets termed interface 2 and interface 3. All are targetable using small molecules [16]. The central pocket is predominantly occupied by palmitate and myristate lipids that are covalently linked to TEADs via acylation to the cysteine in the central pocket. When acylated, the central pocket is inaccessible to small molecules. However, ATA functioned as a disruptor even when we used acylated TEAD. Therefore, we predict that ATA binds on the TEAD surface. We further delineated the ATA binding site and showed, using a VGLL1-TEAD FP assay, that ATA does not bind to interface 2, but likely binds to the interface 3 pocket on the TEAD surface. As the interface 3 residues are identical in all TEADs, we predict that ATA will effectively interact with all four TEAD family members.

We leveraged the activity of the oncogenic gain-of-function TC fusion protein to determine whether ATA can suppress TC induced cellular transformation in cells. Similar to wild-type TAZ, TC is a transcriptional coactivator and maintains the TEAD binding motif present in the TAZ protein. Further, we previously reported that the TC [S51A] mutation that abrogates the TC-TEAD interaction, completely inhibited cellular transformation [23, 34]. Consistent with its function as a TAZ-TEAD disruptor, ATA reduced TC-mediated enhancement of TEAD reporter activity and expression of TC-TEAD regulated genes, as assessed by RNA-Seq and qPCR. ATA also inhibited TC-induced soft agar colony growth of NIH3T3 cells, which we

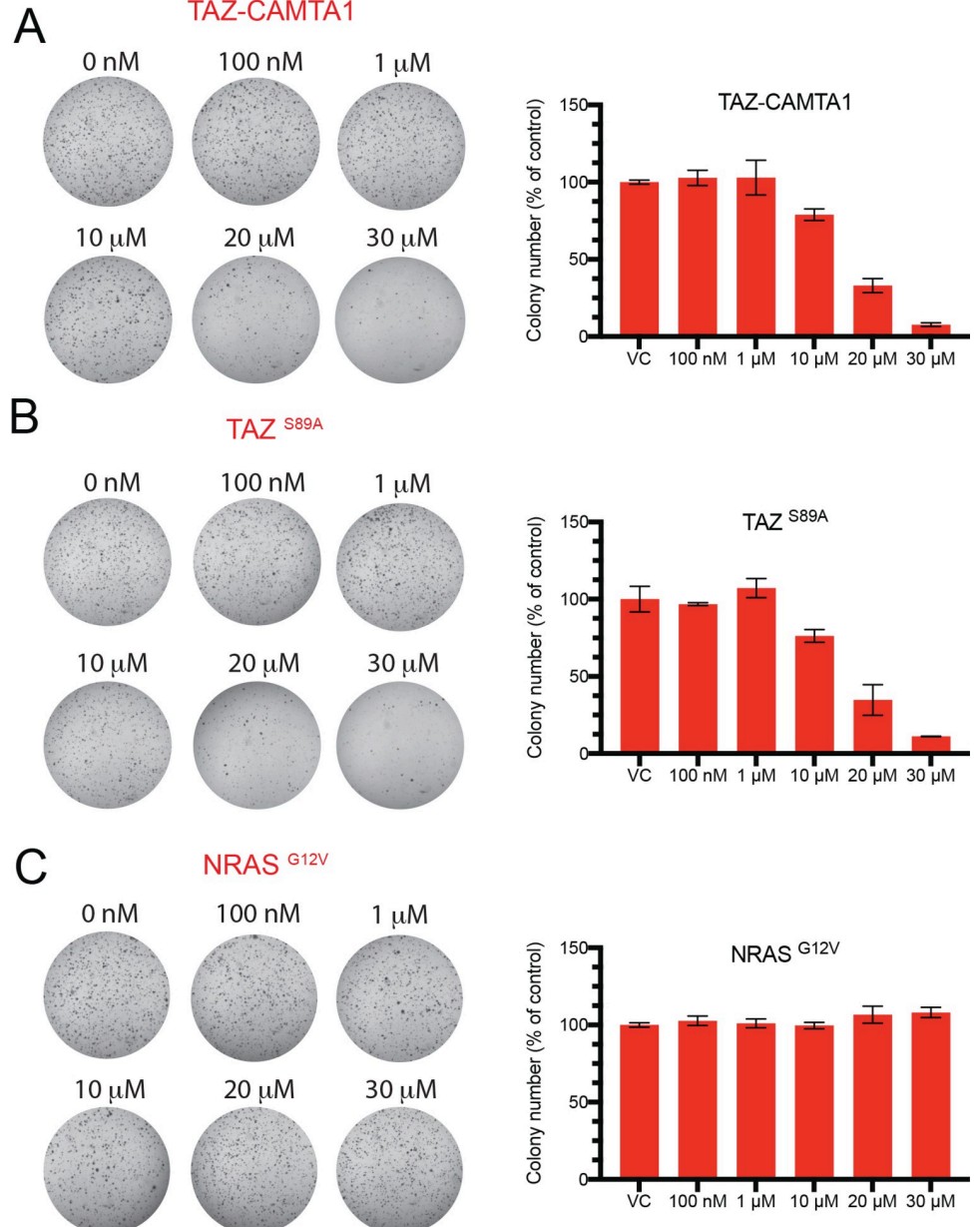

**Fig 4. ATA inhibits soft agar colony growth that is dependent on TC/TAZ-TEAD activity.** Bright-field images and the corresponding quantification of soft agar colonies determined as percent normalized to vehicle controls (VC). Cells were treated with the indicated doses of ATA in (A) NIH3T3 cells expressing TAZ-CAMTA1 (B) NIH3T3 expressing TAZ [S89A] and (C) NIH3T3 cells expressing NRAS [G12V]. Data represent results of two independent experiments. Data are presented as mean (n = 3) and error bars represent the standard deviation of the mean.

have previously shown to be a remarkable readout of the TC-TEAD interaction and the complex's transcriptional activity [23]. The soft agar colony growth induced by TAZ [S89A] is similarly inhibited by ATA in a dose-dependent fashion. The reduction in the number of colonies in soft agar by ATA is not due to cytotoxicity because NRAS-transformed NIH3T3 colonies were insensitive to ATA addition.

ATA can inhibit a diverse group of proteins. For instance, ATA was identified as an inhibitor of cell-free protein synthesis through its ability to disrupt protein-RNA interactions [37].

The utility of ATA as an antithrombotic drug has also been investigated as it acts as a disruptor of von Willebrand factor (vWF)—platelet glycoprotein 1b (GP1b) interaction [38]. ATA has also been shown to act as an inhibitor of the interaction between tumor necrosis factor-like weak inducer of apoptosis (TWEAK) and fibroblast growth factor-inducible 14 (Fn14) [39]. As ATA can disrupt other bimolecular complexes, it requires modification to allow it to act as a specific inhibitor of the TAZ-TEAD complex. Toward this end, we synthesized a small library of ATA analogs that contained specific alterations in the ATA scaffold. These analogs were not as potent as ATA and therefore were not used in further studies. However, we were not comprehensive with our analog synthesis, so we are continuing work to modify ATA to improve its selectivity for TEAD.

The majority of the identified small molecule YAP/TAZ-TEAD disruptors bind to the central pocket of TEAD and therefore have an allosteric mode of action [16, 40]. Some of the molecules, such as MGH-CP1 and others, bind non-covalently, whereas others like TEAD-347 and MYF-01-037 are covalent ligands [41–44]. Central pocket-binding ligands identified by Vivace Therapeutics and Ikena Oncology have already entered clinical trials (NCT04665206, NCT05228015), and we will know their efficacy in humans soon. The other small molecule YAP-TEAD disruptor that had entered clinical trial is the IAG933 from Novartis (NCT04857372), but its mechanism of action has not been disclosed.

In contrast to allosteric disruptors, canonical disruptors bind to the interfaces on the TEAD surface and interfere with YAP/TAZ or TC binding to TEAD. Thus, they have a simple and straightforward mechanism of action, and they do not need to outcompete a ligand, unlike compounds that would bind the central pocket. As the surface pockets on TEADs are shallow, identifying an effective small molecule disruptor of the YAP/TAZ-TEAD interaction has been difficult. Peptides nevertheless can bind the surface pockets and effectively act as disruptors of the YAP-TEAD interaction. Cysteine-dense peptide TB2G1 is a disruptor that acts at interface 2 [45], whereas peptide 17, a cyclic YAP peptide [33], and peptides 9,10, two linear YAP peptides [46], act as disruptors by binding to TEAD at interface 3. However, these peptides have poor cell-penetrating abilities. Small molecules, on the other hand, can be more effectively designed to penetrate cells.

Tri-substituted pyrazoles are small molecule ligands that bind TEAD interface 2 and can be improved to act as effective YAP-TEAD disruptors [47]. At the interface 3 site, a small molecule ligand with a dioxo-benzoisothiazole scaffold has been reported to act as a canonical disruptor of YAP-TEAD interaction [48]. We have previously shown that flufenamic acid has a secondary binding site at interface 3 [18]. At this site, flufenamic acid only weakly interacts with TEAD, so it is not an effective disruptor. Therefore, ATA is another small molecule that can act as an effective disruptor by binding at interface 3. Identification of the small molecule ATA as a canonical disruptor of the TAZ-TEAD interaction is an important development and it opens up new opportunities for targeting the undruggable YAP/TAZ-TEAD interaction.

## Supporting information

**S1 Fig. TAZ-TEAD AlphaLISA assay development.** (A) Coomassie stain (CS) and western blots of full-length purified TAZ and TEAD that were used in the TAZ-TEAD AlphaLISA assay. (B) Various ratios of full-length TAZ and TEAD were used to identify the concentrations that produce the maximal alpha signal, seen at the "hook" point. (C) Distribution plot of the bioactive screen, compounds that showed greater than 70% inhibition were shortlisted as hits. (D) Dose-response curves of ATA and other ATA analogs that show inhibition in the TAZ-TEAD AlphaLISA screen.
(TIF)

**S2 Fig. TAZ-TEAD fluorescence polarization (FP) assay.** The effectiveness of ATA was compared with that of verteporfin and peptide 17. Both verteporfin and peptide 17 were identified as disruptors of the TEAD complex. Verteporfin did not display an effect in this assay.
(TIF)

**S3 Fig. Lactate dehydrogenase (LDH) cytotoxicity assay.** NIH3T3 and HEK293 cells were treated with the indicated concentrations of ATA and the cytotoxicity was evaluated by calculating the amount of intracellular LDH released into the media. LDH absorbance after complete cell lysis is considered as 100% cytotoxicity and the absorbance after ATA treatment was normalized accordingly.
(TIF)

**S1 Table. Structures of ATA analogs and their IC$_{50}$ values as determined from the TAZ-TEAD AlphaLISA assay.**
(ZIP)

**S2 Table. Differentially expressed genes.**
(XLSX)

**S1 Raw images.**
(TIF)

## Acknowledgments

We are indebted to Drew J. Adams for his advice on the TAZ-TEAD AlphaLISA screen development. We thank the molecular biotechnology core of the Lerner Research Institute for their help with surface plasmon resonance measurements. We thank the Cleveland Clinic Center for Therapeutics Discovery (C3TD) for selecting the ATA analogs. We acknowledge the effort of Dr. Cassandra Talerico for the critical reading of this manuscript.

## Author Contributions

**Conceptualization:** Brian P. Rubin.

**Data curation:** Kepeng Che, Caleb N. Seavey, Yuriy Fedorov, Anton A. Komar, Shuang Ma.

**Formal analysis:** Kepeng Che, Ajaybabu V. Pobbati, Caleb N. Seavey, Brian P. Rubin.

**Funding acquisition:** Brian P. Rubin.

**Investigation:** Ajaybabu V. Pobbati, Caleb N. Seavey.

**Methodology:** Kepeng Che, Ajaybabu V. Pobbati, Ashley Burtscher.

**Project administration:** Brian P. Rubin.

**Resources:** Brian P. Rubin.

**Validation:** Kepeng Che, Ajaybabu V. Pobbati.

**Writing – original draft:** Ajaybabu V. Pobbati.

**Writing – review & editing:** Ajaybabu V. Pobbati, Caleb N. Seavey, Brian P. Rubin.

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
