## [Decision Letter · Decision Letter 0]

24 Jan 2022

PONE-D-21-39096Aurintricarboxylic acid is a canonical disruptor of the TAZ-TEAD transcriptional complexPLOS ONE

Dear Dr. Rubin,

Thank you for submitting your manuscript to PLOS ONE. After careful consideration, we feel that it has merit but does not fully meet PLOS ONE’s publication criteria as it currently stands. Therefore, we invite you to submit a substantially revised version of the manuscript that addresses the points raised during the review process. Specifically, the reviewer has identified several major and minor issues that require further attention before the manuscript can be published. Please carefully address all these issues in the revised manuscript.

We look forward to receiving your revised manuscript.

Kind regards,

Jinsong Zhang

Academic Editor

PLOS ONE

Journal Requirements:

"HTS was supported by the Small Molecule Drug Development shared resource of the Case Comprehensive Cancer Center (P30CA043703). We are indebted to Drew J. Adams for his advice with the TAZ-TEAD AlphaLISA screen development. We thank the molecular biotechnology core of the Lerner Research Institute for their help with surface plasmon resonance measurements. C.N.S. is supported in part through the Crile Research Fellowship. This research is supported by generous funding from the EHE Foundation, The EHE Rare Cancer Charity (UK), The EHE Rare Cancer Foundation Australia and the Margie and Robert E. Petersen Foundation to B.P.R. We thank the Cleveland Clinic Center for Therapeutics Discovery (C3TD) for their help with the selection of ATA analogs. We acknowledge the effort of Dr. Cassandra Talerico for critical reading of this manuscript."

We note that you have provided funding information. However, funding information should not appear in the Acknowledgments section or other areas of your manuscript. We will only publish funding information present in the Funding Statement section of the online submission form. 

"This research is supported by generous funding from the EHE Foundation, The EHE Rare Cancer Charity (UK), The EHE Rare Cancer Foundation Australia and the Margie and Robert E. Petersen Foundation awarded to B.P.R. These are Foundation grants and they do not carry a grant number. 

Screening campaign was supported by the Small Molecule Drug Development shared resource of the Case Comprehensive Cancer Center (P30CA043703). 

C.N.S. is supported in part by the Crile Research Fellowship.

Reviewers' comments:

Reviewer's Responses to Questions

**Comments to the Author**

1. Is the manuscript technically sound, and do the data support the conclusions?

Reviewer #1: Yes

2. Has the statistical analysis been performed appropriately and rigorously? 

Reviewer #1: Yes

3. Have the authors made all data underlying the findings in their manuscript fully available?

Reviewer #1: Yes

4. Is the manuscript presented in an intelligible fashion and written in standard English?

Reviewer #1: Yes

5. Review Comments to the Author

Reviewer #1: In this manuscript, the authors found aurintricarboxylic acid (ATA) can disrupt the complex formation between TAZ and TEAD4 when they identified ATA as a result for high-throughput screening of compound library by the method of AlphaLISA assay. They confirm that ATA inhibited tumor growth in addition to TAZ/YAP/TEAD-mediated transcriptional assays. Since it is not mentioned that ATA possesses any side-effects for our bodies, ATA might be a novel seed to suppress TAZ/YAP/TEAD-induced tumorigenicity. However, there are a few concerns before publication.

(Major concerns)

1. If the authors want to show that ATA specifically inhibit TAZ/YAP/TEAD-induced tumorigenicity, the tumor cells with or without TEAD must be treated with ATA although they introduced an active TAZ in NIH3T3 cells. If it is difficult, an alternative approach is to use tumor cells with or without TAZ/YAP.

2. They should try to compare the inhibitory activities between ATA and known TAZ/YAP/TEAD inhibitor(s) such as verteporfin.

3. In vivo experiment with ATA should be tried.

4. Does ATA interact with either YAP or TAZ?

(Minor concerns)

1. In Introduction or Discussion, they should describe YAP/TAZ or TEAD inhibitors that have been published. AND Discuss about comparison between ATA and known inhibitors.

2. Discuss more about functional moieties of ATA and its necessity.

6. PLOS authors have the option to publish the peer review history of their article (what does this mean?). If published, this will include your full peer review and any attached files.

Reviewer #1: **Yes: **Susumu Itoh

---

## [Author Response · Author response to Decision Letter 0]

9 Mar 2022

Dear Dr. Zhang,

We appreciate the critical review of our manuscript entitled “Aurintricarboxylic acid is a canonical disruptor of the TAZ-TEAD transcriptional complex,” which have improved our study. We here submit a revised version of the manuscript in accordance with the journal guidelines and after addressing the reviewer’s major and minor concerns - a point-by-point response is provided below.

Further, in order to comply with PLOS ONE publication criteria, we have formatted the manuscript as per the journal guidelines. We have removed funding information details from the acknowledgments section. Please include our funding statement provided below in the funding statement section of the online submission form.

“This research is supported by generous funding from The EHE Foundation, The EHE Rare Cancer Charity (UK), The EHE Rare Cancer Foundation Australia and the Margie and Robert E. Petersen Foundation. These grants were awarded to B.P.R. These foundation grants do not carry a grant number. The high throughput screening campaign was supported by the Small Molecule Drug Development shared resource of the Case Comprehensive Cancer Center (P30CA043703). C.N.S. is supported in part by the Crile Research Fellowship. The funders had no role in study design, data collection and analysis, decision to publish, or preparation of the manuscript.” 

All relevant data are provided in the manuscript main figures and in the supporting information. Full blot/gel images are also provided as supporting information.

Point-by-point response to the reviewer

Reviewer #1: In this manuscript, the authors found aurintricarboxylic acid (ATA) can disrupt the complex formation between TAZ and TEAD4 when they identified ATA as a result for high-throughput screening of compound library by the method of AlphaLISA assay. They confirm that ATA inhibited tumor growth in addition to TAZ/YAP/TEAD-mediated transcriptional assays. Since it is not mentioned that ATA possesses any side-effects for our bodies, ATA might be a novel seed to suppress TAZ/YAP/TEAD-induced tumorigenicity. However, there are a few concerns before publication.

We thank the reviewer for appreciating the merit of our study. ATA is a novel molecule that possess the canonical mechanism of action of a PPI inhibitor. It binds to TEAD-TAZ interaction interface and disrupts the formation of the TAZ-TEAD complex. As the reviewer mentioned, we do believe that the identification of ATA as a hit needs to be disseminated to allow further investigation that may facilitate development of potent ATA analogs for targeting oncogenicity driven by YAP/TAZ-TEAD. 

(Major concerns)

1. If the authors want to show that ATA specifically inhibit TAZ/YAP/TEAD-induced tumorigenicity, the tumor cells with or without TEAD must be treated with ATA although they introduced an active TAZ in NIH3T3 cells. If it is difficult, an alternative approach is to use tumor cells with or without TAZ/YAP.

Indeed, this would be a great experiment to perform to knockout YAP/TAZ or TEAD in NIH3T3 cells and evaluate the effects of ATA. However, in this model system, the endogenous levels of YAP/TAZ or TEAD are not sufficient to transform these cells. Therefore, the phenotype of wildtype versus YAP/TAZ or TEAD knockout cells would be the same, which makes it impossible to test the selectivity of ATA. The YAP/TAZ-TEAD transcriptional complex needs to be dysregulated so that it evades endogenous cellular control mechanisms. Human cancer cells commonly evade control by increasing YAP/TAZ levels or generate constitutively active fusion proteins of YAP or TAZ. Therefore, we mimicked this scenario through exogenous expression of active TAZ (TAZ S89A), or TAZ-CAMTA1 (TC) fusion protein. In our previously published manuscript (Tanas M, et. al., Oncogene, 2016), several control experiments were performed to validate this cell-based model. For instance, we mutated a single key residue in TC (S51A) that abrogates TEAD interaction and demonstrated that the mutant TC loses its transforming ability, reinforcing that these cells are transformed through TC/TEAD interaction.

Further, we show in Fig 4 in this manuscript that ATA specifically inhibits the colony growth of cells transformed by the expression of TAZ S89A or TC, but not by cells transformed through NRAS G12V expression. In NRAS-transformed cells, the tumorigenicity occurs through mechanisms that are independent of TAZ/TEAD activity and ATA has no effect on these cells, suggesting that ATA selectively inhibits TAZ/TEAD activity. 

2. They should try to compare the inhibitory activities between ATA and known TAZ/YAP/TEAD inhibitor(s) such as verteporfin.

As per the reviewer’s suggestion, we performed the experiment using the TAZ-TEAD FP assay, in which we compared the activity of ATA with peptide 17, a known PPI inhibitor that acts by binding to the interface 3 region similar to ATA (Zhang., Z et. al., ACS Medicinal Chemistry Letters, 2014). We also measured the activity of verteporfin in this assay. Both ATA and peptide 17 are active but verteporfin had no effect. 

It is becoming increasingly clear that verteporfin may not be a direct PPI inhibitor but may act upstream by sequestering YAP/TAZ in the cytoplasm through increasing the levels of 14-3-3σ (Wang et. al., Am J Cancer Res, 2016). Accordingly, we did not observe a direct PPI inhibition when verteporfin was tested. 

The results of the experiment suggested by the reviewer are shown below and are now included in new Supplementary Figure 2 in the revised version of the manuscript.

3. In vivo experiment with ATA should be tried.

A suitable in vivo experiment can only be performed after furthering ATA through a hit-to-lead and lead optimization pipeline. This usually involves extensive chemistry and evaluation of DMPK in vitro and in animal models. We have to ensure that the drug can stably reach the required concentration for a reasonable amount of time in vivo and is not cleared. This will allow us to confirm that the potential lack of any effect is not due to lack of bioavailability or stability. These studies are beyond the scope of this current study.

4. Does ATA interact with either YAP or TAZ?

We have performed a ThermoFluor assay that measures protein stability using full-length proteins and after ATA addition. A significant increase in melting temperature was observed upon ATA addition to TEAD than when ATA was added to TAZ. These results are shown in Fig 1E and 1F. Therefore, the preferred binding partner of ATA is TEAD, and ATA may only weakly interact with YAP or TAZ.

(Minor concerns)

1. In Introduction or Discussion, they should describe YAP/TAZ or TEAD inhibitors that have been published. AND Discuss about comparison between ATA and known inhibitors.

As per the reviewer’s suggestion, we have now expanded our discussion and compared the attributes of various disruptors and ATA.

2. Discuss more about functional moieties of ATA and its necessity.

We thank the reviewer for this comment. From our SAR, we observed that the salicylate group is important for activity because compounds lacking it were ineffective. For more potent inhibition, two salicylate groups and an additional aryl group may be required. To emphasize these, we have rephrased the Results, section “Synthesis of ATA analogs” in the following way:

“A minimum pharmacophore study was conducted to identify key functionalities of ATA that disrupt the TAZ-TEAD interaction. In total, 16 ATA analogs were tested. Compounds 1 and 2, which retained three phenyl substituents like ATA but lacked the salicylate group were not appreciably potent (S1 Table). Therefore, the salicylate group is important for disrupting the TAZ-TEAD interaction. To further probe the requirement of the salicylate group, we assayed five analogs (compounds 3, 4, 5, 6, and 7) that had two aryl substituents. Compound 4, which featured two salicylate moieties, was the most potent, with an IC50 of 8 μM in the TAZ-TEAD AlphaLISA assay (S1 Table). Although compound 4 displayed activity, it was not as effective as ATA, which had a third aryl substituent in addition to the two salicylate moieties. Therefore, three aryl substitutions and two salicylate moieties may be important for effective disruption of TAZ-TEAD interaction. Additionally, no analogs with one or zero aryl substituents displayed inhibition. Next, we evaluated five derivatives of compound 4 by introducing mono-, di- or gem-dimethyl groups at the methylene linker, or capped hydroxyls with methyl groups, and assessed their potencies (compounds 12-16, S1 Table). Compound 14 was estimated to be most potent of the 16 ATA analogs with an IC50 of 4 μM. This compound had a mono-methyl group at the methylene linker. We also observed that capping the hydroxyls with methyl groups as in compound 16 abolished the activity. Nevertheless, all the tested analogs were substantially less potent than ATA, which had an IC50 of 35 nM (S1D Fig). As a result, we further characterized ATA and not its analogs in subsequent studies.”

Sincerely,

Brian P Rubin

---

## [Editor Report · Decision Letter 1]

15 Mar 2022

Aurintricarboxylic acid is a canonical disruptor of the TAZ-TEAD transcriptional complex

PONE-D-21-39096R1

Dear Dr. Rubin,

We’re pleased to inform you that your manuscript has been judged scientifically suitable for publication and will be formally accepted for publication once it meets all outstanding technical requirements.

Kind regards,

Jinsong Zhang

Academic Editor

PLOS ONE
---

## [Editor Report · Acceptance letter]

22 Mar 2022

PONE-D-21-39096R1 

Aurintricarboxylic acid is a canonical disruptor of the TAZ-TEAD transcriptional complex 

Dear Dr. Rubin:

I'm pleased to inform you that your manuscript has been deemed suitable for publication in PLOS ONE. Congratulations! Your manuscript is now with our production department. 

Kind regards, 

on behalf of

Dr. Jinsong Zhang 

Academic Editor

PLOS ONE